# VideoHandles: Editing 3D Object Compositions in Videos Using Video Generative Priors

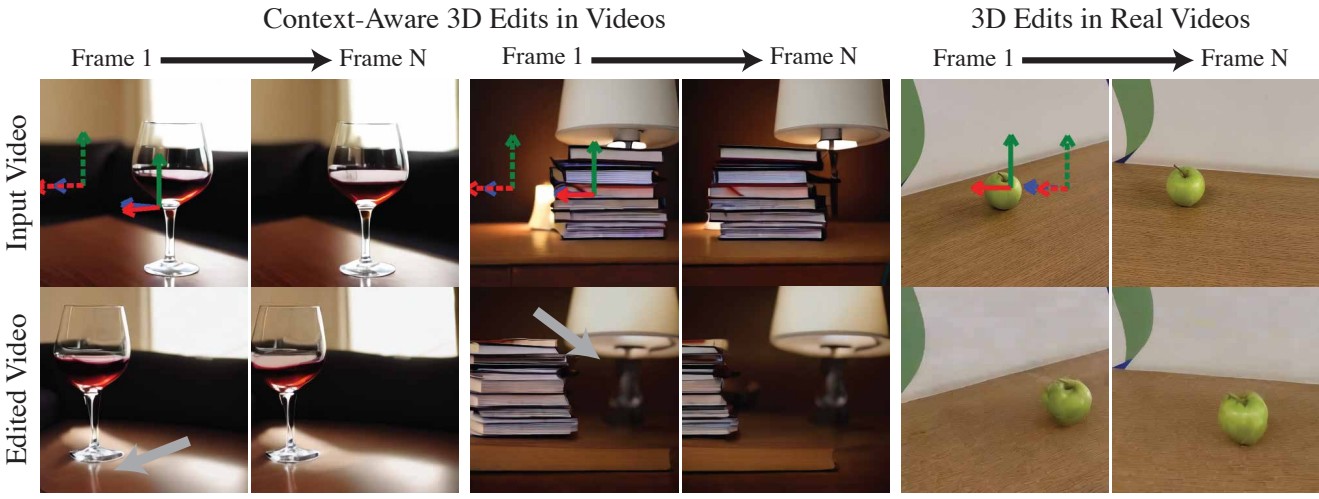

Figure 1. **VideoHandles edits 3D object composition in videos of static scenes.** Solid axes represent the original 3D position and dotted axes the user-provided target position. The edit plausibly updates effects like the reflection of the wine glass and handles disocclusions like the lamp behind the book pile that is exposed by the edit. In addition to generated videos, we can also edit real (non-generated) videos by inverting the video into its corresponding latent, as shown on the right.

## Abstract

*Generative methods for image and video editing use generative models as priors to perform edits despite incomplete information, such as changing the composition of 3D objects shown in a single image. Recent methods have shown promising composition editing results in the image setting, but in the video setting, editing methods have focused on editing object's appearance and motion, or camera motion, and as a result, methods to edit object composition in videos are still missing. We propose VideoHandles as a method for editing 3D object compositions in videos of static scenes with camera motion. Our approach allows editing the 3D position of a 3D object across all frames of a video in a temporally consistent manner. This is achieved by lifting intermediate features of a generative model to a 3D reconstruction that is shared between all frames, editing the reconstruction, and projecting the features on the edited recon-*
*struction back to each frame. To the best of our knowledge, this is the first generative approach to edit object compositions in videos. Our approach is simple and training-free, while outperforming state-of-the-art image editing baselines.*

## 1. Introduction

Diffusion models and flow-based models are currently the standard for high-quality text-to-image generation. Text-to-video diffusion/flow-based models lag behind in quality, but have recently seen big improvements. The prevalent text-based control is easy to use, but impractical for some types of edits, such as edits of the object composition in a scene: specifying the position of an object with text is inaccurate and iterative editing workflows are not supported. Several recent methods address this issue in the image domain by proposing different types of iterative image editing

methods. These either focus on editing the appearance of objects [4, 13, 46], or their spatial composition [1, 3, 28]. In the video domain, current methods support editing only the appearance [6, 22] while lacking methods to edit spatial object compositions, for example, editing the 3D position of objects in generated videos, as shown in Figure 1. Editing the object composition in a video introduces several challenges: a *plausible* editing output requires generating details such as shadows and lighting that may have changed due to the edited composition; furthermore, the edited video needs to *preserve the identity* of the original objects and should *adhere to an edit control* manipulated by the user. Finally, the edit needs to be applied to all video frames in a *temporally consistent* manner.

We propose *VideoHandles* as a generative approach to edit the object composition in a video of a static scene. Our approach allows editing the 3D position of a 3D object in a video, resulting in a plausible, temporally consistent edit that preserves the identity of the original object. To the best of our knowledge, ours is the first generative approach that allows editing the object composition in a video. Given a pretrained flow-based video generative model, we present a novel method to edit the intermediate features from the generative model's network in a temporally consistent manner. Specifically, we lift the intermediate features of each frame to a common 3D reconstruction, effectively treating them as latent textures. We then edit the 3D location of an object using 3D translations or rotations, and project the features back to their corresponding frames. We use such projected features as guidance during the generative process to create a plausible edited video. Editing of real (non-generated) videos is supported by first inverting them into the random noise. Our approach is simple and does not require any training or finetuning that risks biasing the distribution of the generative model.

We evaluate our method on several generated and captured videos. As there are no existing methods that are specialized to editing the 3D object composition in videos, we compare to several image editing baselines that can be applied in a per-frame manner. We evaluate the results in terms of plausibility, temporal consistency, identity preservation, and adherence to the target edit. In addition to a large number of qualitative comparisons, we also conduct a user study. The users have a clear preference for our method in terms of plausibility and temporal consistency, while our method is at least on par or slightly better than image editing baselines in terms of identity preservation and edit adherence. Finally, we perform a quantitative evaluation which confirms these findings.

We summarize our contributions as follows:

- We introduce a zero-shot method for editing object composition in videos using video generative priors, for the first time to our knowledge.

- For the feature-based generative editing process, we describe the optimal approach for feature extraction from the video generative model network (Section 4.2).
- We also demonstrate that self-attention-map-based weighting (Section 4.4) and null-text prediction in the foreground region (Section 4.5) further improve the editing quality.
- We demonstrate the effectiveness of our method with both generated and real videos.

## 2. Related Work

In the context of diffusion/flow-based generative models, several methods have been proposed for image and video editing that can be roughly grouped by the type of edits they perform.

**Image Appearance Editing.** There has been a series of work that focus on manipulating intermediate features or attention maps of pre-trained image diffusion models to edit the appearance of objects within an image [4, 5, 10, 13, 41, 46] in a zero-shot setting. While effective, such methods often do not focus on editing the composition of objects, which requires control over object positions and strict identity preservation. To tackle identity preservation, various customization approaches have been proposed that enable the generation of images of a particular object or subject in different compositions. However, such methods do not provide edit controls and typically require finetuning of the base model [18, 32]. The prior of image diffusion models has been further utilized to enable editing of 3D static scenes represented as 3D neural assets via iterative optimization approaches [16, 17, 29]. These methods, however, also focus on changing the appearance of objects, rather than our goal of composition editing.

**Image Composition Editing.** Several recent methods aim at editing the composition of objects in an image [1–3, 7–9, 25, 28, 47, 49]. Another line of work aims at inserting an object from a source image into a new target image [38, 39, 45], which can be repurposed as image editing tools by using the same image as source and target. Another popular editing workflow provides control points that can be dragged by a user to deform objects or edit 2D object positions [21, 27, 34–36]. Additionally, a few more general image editing methods have been proposed that can be used for either image appearance editing or image composition editing [24, 48]. All of these methods can be applied to videos by separately editing each frame, but this loses temporal consistency, as we show in our experiments in Section 5. Most related to our work is Diffusion Handles [28] which inspired our approach of editing intermediate features using a 3D reconstruction. We show how to modify this approach so it can be applied to non-depth-conditioned video priors, including which features to pick, which 3D

reconstruction method to use, how to avoid artifacts from hard object masks, and how to effectively remove the original object from the edited video.

**Video Appearance Editing.** With the increasing quality of video generators, various works have focused on editing the appearance of objects within videos. A key issue these works aim to tackle is to maintain temporal consistency between frames while changing the appearance. To address this, some works [6, 7, 31] have proposed techniques to maintain consistency using only image diffusion models, while others [14, 15] have leveraged priors from video diffusion models to tackle this challenge. While successful in preserving temporal consistency, these approaches are limited to appearance changes, and no prior work addresses changing object compositions in videos.

**Video Motion Control.** Recently, another line of work in the video domain focuses on controlling the motion of objects or cameras during generation [19, 33, 42, 44]. Although these methods allow specifying how a particular object should move in a video, they are designed for generation rather than editing tasks, thus they do not allow modifying the compositions of static object arrangements in videos. Moreover, unlike these approaches that require training on task-specific datasets to learn motion control, *VideoHandles* is training-free.

## 3. Preliminary: Flow-Based Latent Video Model

In this section, we briefly discuss the video prior we use in our experiments, which is the flow-based latent video model, OpenSora [51].

**Flow-Based Generative Model.** Similar to diffusion models [12, 37], flow-based generative models [20, 23] model high-dimensional data distributions through a learned iterative process. Given a data sample $\boldsymbol{Z}_1 \sim p_{\text{data}}$ and random noise $\boldsymbol{Z}_0 \sim \mathcal{N}(\boldsymbol{0}, \boldsymbol{I})$, a linear trajectory is defined as $\boldsymbol{Z}_t = t\boldsymbol{Z}_1 + (1-t)\boldsymbol{Z}_0$. Based on the linear trajectory, a veloicty prediction network $v_\theta$ is trained to estimate the derivative $d\boldsymbol{Z}_t/dt$:

$$v_\theta(\boldsymbol{Z}_t, t, y) \approx \frac{d}{dt}\boldsymbol{Z}_t = \boldsymbol{Z}_1 - \boldsymbol{Z}_0, \qquad (1)$$

where $y$ encodes the text prompt corresponding to $\boldsymbol{Z}_1$. Given a trained velocity prediction network $v_\theta$, a new data sample can be generated through the generative process, starting from $\boldsymbol{Z}_0$:

$$\boldsymbol{Z}_{t+\Delta t} = \boldsymbol{Z}_t + \Delta t \cdot v_\theta^\omega(\boldsymbol{Z}_t, t, y), \qquad (2)$$

where $v_\theta^\omega(\boldsymbol{Z}_t, t, y) = v_\theta(\boldsymbol{Z}_t, t, \varnothing) + \omega(v_\theta(\boldsymbol{Z}_t, t, y) - v_\theta(\boldsymbol{Z}_t, t, \varnothing))$ denotes a prediction using classifier-free

guidance [11] with null-text embedding $\varnothing$ and guidance scale $\omega$. The step size $\Delta t$ can be chosen at inference time to balance quality with speed.

**DiT-Based Architecture for Latent Video Model.** A video $\boldsymbol{X} \in \mathbb{R}^{n \times h \times w \times 3}$ with $n$ frames is encoded into a latent representation $\boldsymbol{Z}_1 \in \mathbb{R}^{M \times H \times W \times D}$ by a pre-trained encoder, where all dimensions except the feature dimension $D$ are reduced. Each pixel of the latent representation encodes a spatio-temporal patch of $\boldsymbol{X}$. The velocity prediction network $v_\theta$ is implemented as a DiT [30] that operates on this latent representation, with alternating blocks of spatial self-attention, temporal self-attention, and cross-attention to the text prompt. A total of 24 blocks of each type are used. A latent sampled from the generative process is decoded by a pre-trained decoder to produce a video sample.

## 4. VideoHandles: A 3D-Aware Video Editing Method

Consider a static input video $\boldsymbol{X}_{\text{src}} \in \mathbb{R}^{n \times h \times w \times 3}$, where objects remain stationary and only the camera moves. Our goal is to apply a 3D transformation to an object selected by the user in the first frame while preserving the identity of the input video, realism, and temporal consistency. See Figure 2 for an architecture overview.

To ensure that transformations in each frame of a video align with those in other frames, we define a 3D space in which a point cloud $\boldsymbol{P}_{\text{src}} = \{\mathbf{p}^{(j)}\}_{j=1}^{J}$ represents the 3D scene in the video with a shared coordinate system across all frames. A transformation is performed in this shared 3D space, denoted by $\mathcal{T} : \mathbb{R}^3 \rightarrow \mathbb{R}^3$, with each input frame $\mathbf{x}_{\text{src}}^{(i)}$ modeled as a 2D rendering of $\boldsymbol{P}_{\text{src}}$ from the $i$-th view. Specifically, we reconstruct $\boldsymbol{P}_{\text{src}}$ and estimate a camera pose for each frame from $\boldsymbol{X}_{\text{src}}$ using DUST3R [43]. By leveraging the reconstructed 3D scene from $\boldsymbol{X}_{\text{src}}$, we define a 3D-aware warping function in the 2D space of each frame.

However, due to inaccuracies in warping caused by errors in reconstructing the 3D scene, directly warping pixel colors often leads to unrealistic videos. Moreover, this approach fails to appropriately adjust the video according to the 3D scene and the transformed object, such as new shadows, reflections, and relighting effects. Therefore, inspired by Diffusion Handles [28], we perform warping in the *feature* space of a pre-trained video generative model and use the warped features as guidance during the generative process. This ensures that the generative prior of the video model adapts the scene with appropriate context changes according to the new object composition while maintaining temporal consistency.

In the following sections, we first introduce how to compute the warping function for each 2D frame based on the transformation of an object in the 3D scene (Section 4.1).

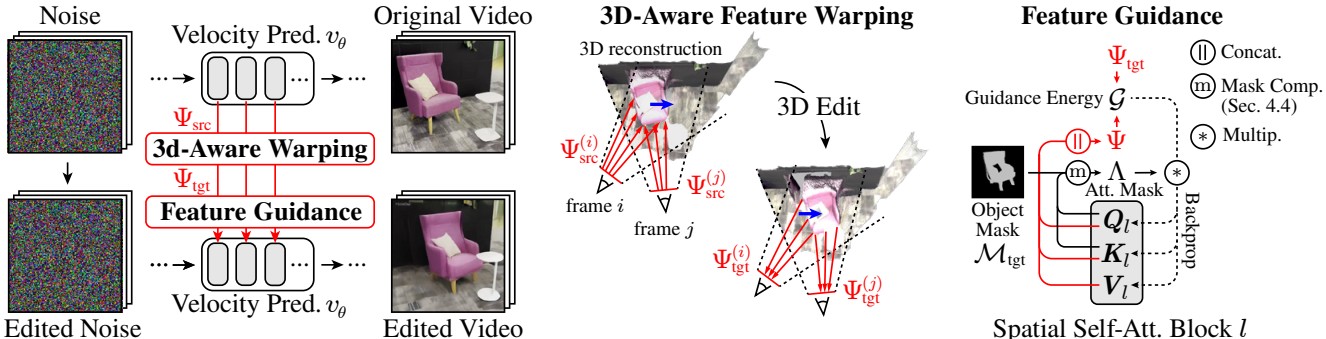

Figure 2. **VideoHandles Architecture.** We use the intermediate features $\Psi_{\text{src}}$ of a video generative model to represent the identity of objects in a source video. Given a 3D transformation of an object, we can use a 3D reconstruction of the scene to warp the intermediate features consistently across frames. Guiding the video generator with these warped features $\Psi_{\text{tgt}}$ gives us a an edited video where the object is transformed, while also maintaining the plausibility of effects like shadows and reflections.

Next, we describe the features of the pretrained flow-based latent video model and how these features are warped (Section 4.2). Lastly, we explain how the warped video model features serve as guidance in the energy-based guided generative process (Section 4.3).

### 4.1. 3D-Aware Warping Function

We first describe how to obtain a 3D-aware warping function in the 2D space of each frame. Given a set of 2D coordinates $\Omega_{H,W} = \{(v, u) \mid v \in [0, H), u \in [0, W)\}$, the connection between the 3D space and the $i$-th 2D frame is established through the *projection function* $f^{(i)} : \mathbb{R}^3 \rightarrow \Omega_{H,W}$, which is defined by the $i$-th camera pose. Let $\mathcal{B}_{\text{src}}^{(1)} : \Omega_{H,W} \rightarrow \{0, 1\}$ denote the 2D binary mask of an object selected by users in the first frame. Based on the 2D object mask in the first frame $\mathcal{B}_{\text{src}}^{(1)}$, we first partition $P_{\text{src}}$, the point cloud reconstructed from the input video $X_{\text{src}}$, as follows:

$$P_f = \{\mathbf{p} \in P_{\text{src}} \mid \mathcal{B}_{\text{src}}^{(1)}(f^{(1)}(\mathbf{p})) = 1\}, \quad (3)$$
$$P_b = P_{\text{src}} \setminus P_f, \quad (4)$$

where $P_f$ consists of points whose projections lie within the 2D masked region defined by $\mathcal{B}_{\text{src}}^{(1)}$, and $P_b$ denotes the remaining points representing the background. By applying a 3D transformation $\mathcal{T}$ to $P_f$ alone, we construct a rough target 3D scene represented as a point cloud:

$$P_{\text{tgt}} = \mathcal{T} P_f \cup P_b. \quad (5)$$

The *lifting function* $g_{\text{src}}^{(i)} : \Omega_{H,W} \rightarrow \mathbb{R}^3$ takes a 2D coordinate $\mathbf{u} = (v, u)$ as input and returns the 3D point in $P_{\text{src}}$ closest to the $i$-th camera from among the points projected close to $\mathbf{u}$:

$$g_{\text{src}}^{(i)}(\mathbf{u}) = \arg\min_{\mathbf{p} \in P_{\text{src},\mathbf{u}}^{(i)}} z^{(i)}(\mathbf{p}), \quad (6)$$

where $P_{\text{src},\mathbf{u}}^{(i)} = \{\mathbf{p} \in P_{\text{src}} \mid \|f^{(i)}(\mathbf{p}) - \mathbf{u}\|_1 < \epsilon\}$ represents the set of 3D points that are projected close to $\mathbf{u}$ and $z^{(i)}(\mathbf{p})$ denotes the distance of point $\mathbf{p}$ from the $i$-th camera. Similarly, $g_{\text{tgt}}^{(i)}(\mathbf{u})$ returns the 3D point in $P_{\text{tgt}}$ closest to the $i$-th camera from among the points projected close to $\mathbf{u}$. Using the functions $g_{\text{src}}^{(i)}$ and $g_{\text{tgt}}^{(i)}$, we define an occlusion-aware foreground point cloud $P_f^{(i)} \subseteq P_f$ for each frame as follows:

$$P_f^{(i)} = \{g_{\text{src}}^{(i)}(\mathbf{u})\} \cap \{\mathcal{T}^{-1} g_{\text{tgt}}^{(i)}(\mathbf{u})\} \cap P_f, \quad (7)$$

where $\mathbf{u} \in \Omega_{H \times W}$. It consists of foreground points that are not occluded by the background either before or after the transformation. Using this 3D information, we compute a 2D warping function $\mathcal{W}^{(i)} : \Omega_{H,W} \rightarrow \Omega_{H,W}$ as follows:

$$\mathcal{W}^{(i)}(\mathbf{u}) = \begin{cases} f^{(i)}\big(\mathcal{T}^{-1} g_{\text{tgt}}^{(i)}(\mathbf{u})\big), & \text{if } g_{\text{tgt}}^{(i)}(\mathbf{u}) \in P_f^{(i)} \\ \mathbf{u}, & \text{otherwise.} \end{cases} \quad (8)$$

This warping function gives us the corresponding coordinate in the source image for any coordinate in the target image. All coordinates that do not project to the edited foreground point cloud remain unchanged. We denote warping a 2D signal $\mathcal{X} : \Omega_{H,W} \rightarrow \mathbb{R}^C$ as $(\mathcal{W}^{(i)} * \mathcal{X})(\mathbf{u}) := \mathcal{X}(\mathcal{W}^{(i)}(\mathbf{u}))$. Similarly, we denote its application to a tensor $X \in \mathbb{R}^{\cdots \times H \times W \times \cdots}$ as $\mathcal{W}^{(i)} * X$. Here $H$ and $W$ are the two spatial tensor dimensions that the warping is applied to and the ellipses denote arbitrary additional dimensions. The tensor is sampled at non-integer coordinates using linear interpolation.

As we will show in our evaluation, directly warping RGB frames results in a noisy video, due to inaccuracies in camera predictions and 3D reconstructions, and since this direct warping does not update effects like reflections and shadows that may have changed due to the edit. Therefore, we propose warping the *features* instead of the frames in the

video and synthesizing the edited video through a generative process that guides the features of the edited video to match the warped features. In the next section, we introduce our choice of features for the guided generative process.

## 4.2. Warping Video Features

In this section, we describe our choice of features extracted from OpenSora [51] and explain how these features are warped using the warping function introduced in Section 4.1. The DiT architecture [30] of OpenSora alternates layers that perform spatial self-attention, temporal self-attention, cross-attention to the prompt, and feed-forward computations. Spatial attention operates within each frame, while temporal attention is performed among pixels at the same spatial position across frames. We empirically found that the features from the temporal self-attention layers tend to produce global changes; since each temporal attention layer follows a spatial one, its features tend to affect all pixels in each frame globally. This global spatial context is unsuitable for our local editing tasks, where only the selected object needs to be transformed. Therefore, we use only extract features from the spatial layers for guidance, as these retain more localized information.

Let $\boldsymbol{Q}_l(\boldsymbol{Z}_t), \boldsymbol{K}_l(\boldsymbol{Z}_t), \boldsymbol{V}_l(\boldsymbol{Z}_t) \in \mathbb{R}^{M \times H \times W \times d}$ be the query, key, and value features of the $l$-th self-attention layer extracted from $v_\theta^\omega(\boldsymbol{Z}_t, t, y)$, where $M$ denotes the number of frames and $d$ is the feature dimension. We use their concatenation from all layers as our extracted feature $\Psi$:

$$\Psi(\boldsymbol{Z}_t) = [\boldsymbol{Q}_l(\boldsymbol{Z}_t) \parallel \boldsymbol{K}_l(\boldsymbol{Z}_t) \parallel \boldsymbol{V}_l(\boldsymbol{Z}_t)]_{l=1}^L. \quad (9)$$

Let $\Psi^{(i)}(\boldsymbol{Z}_t) \in \mathbb{R}^{H \times W \times D}$ denote the feature for frame $i$, where $D$ is the total dimensionality of the feature. Applying the previosuly defined warping function, given the latent of the input video $\boldsymbol{Z}_t^{\mathrm{src}}$, its warped feature is defined as $\Psi_{\mathrm{tgt}}^{(i)} := \mathcal{W}^{(i)} * \Psi^{(i)}(\boldsymbol{Z}_t^{\mathrm{src}})$.

## 4.3. Warping-Based Guided Generative Process

To guide the generation process of $\boldsymbol{Z}_t$ with $\Psi_{\mathrm{tgt}}^{(i)}(\boldsymbol{Z}_t)$, we use an energy-guided generative process [8], similar to classifier-free guidance. Given an energy function $\mathcal{G}(\boldsymbol{Z}_t)$, the gradient of $\mathcal{G}$ is injected at each step of the generative process, steering it towards minimizing the energy function:

$$\boldsymbol{Z}_{t+\Delta t} = \boldsymbol{Z}_t + \Delta t \cdot v_\theta^\omega(\boldsymbol{Z}_t, t, y) + \rho \nabla_{\boldsymbol{Z}_t} \mathcal{G}(\boldsymbol{Z}_t), \quad (10)$$

where $\rho$ is a hyperparameter to control the step size of $\nabla_{\boldsymbol{Z}_t} \mathcal{G}$. Below, we describe our specific design of $\mathcal{G}$ to edit object compositions in videos.

**Object transformation energy.** Let $M_{\mathrm{src}}^{(i)}, M_{\mathrm{tgt}}^{(i)} \in \mathbb{R}^{H \times W}$ denote the occlusion-aware 2D masks of the se-

lected object before and after the transformation:

$$\boldsymbol{M}_{\mathrm{src}}^{(i)}(\mathbf{u}) := \begin{cases} 1, \text{ if } \mathbf{u} \in \{f^{(i)}(\mathbf{p}) \mid \mathbf{p} \in \boldsymbol{P}_f^{(i)}\}, \\ 0, \text{ otherwise.} \end{cases}, \quad (11)$$

$$\boldsymbol{M}_{\mathrm{tgt}}^{(i)} := \mathcal{W}^{(i)} * \boldsymbol{M}_{\mathrm{src}}^{(i)}, \quad (12)$$

where $\mathbf{u} \in \Omega_{H \times W}$. Note that $\boldsymbol{M}_{\mathrm{src}}^{(i)}$, which marks the region where the occlusion-aware foreground point cloud $\boldsymbol{P}_f^{(i)}$ is projected, is a subset of the object selection mask $\mathcal{B}_{\mathrm{src}}^{(i)}$ since $\boldsymbol{M}_{\mathrm{src}}^{(i)}$ only includes the object region visible before *and* after the transformation.

To transform the selected object in the video, we define the *object transformation energy* $\mathcal{G}_o(\boldsymbol{Z}_t)$ as follows:

$$\sum_{i=1}^M \left\| \boldsymbol{M}_{\mathrm{tgt}}^{(i)} \odot \left( \Psi_{\mathrm{tgt}}^{(i)} - \Psi^{(i)}(\boldsymbol{Z}_t) \right) \right\|_2^2, \quad (13)$$

where $\odot$ is the element-wise product (broadcasting to additional dimensions where needed). This function measures the discrepancy between the current features $\Psi^{(i)}$ and the target features $\Psi_{\mathrm{tgt}}^{(i)}$ within the region of the edited object $\boldsymbol{M}_{\mathrm{tgt}}^{(i)}$.

**Background preservation energy.** To further preserve background details, we define an additional energy function called the *background preservation energy* $\mathcal{G}_b(\boldsymbol{Z}_t)$ as follows:

$$\left\| \psi_{\mathrm{MHW}} \left( \boldsymbol{M}_b \odot \Psi_{\mathrm{tgt}} \right) - \psi_{\mathrm{MHW}} \left( \boldsymbol{M}_b \odot \Psi(\boldsymbol{Z}_t) \right) \right\|_2^2, \quad (14)$$

where $\psi_{\mathrm{MHW}}$ denotes the average over time and spatial dimensions, and $\boldsymbol{M}_b^{(i)} = \max((1 - \boldsymbol{M}_{\mathrm{src}}^{(i)} - \boldsymbol{M}_{\mathrm{tgt}}^{(i)}, 0)$ is the background mask. This function measures the discrepancy between the sums of the features in the background region. Unlike $\mathcal{G}_o$, $\mathcal{G}_b$ compares only the averages of the features, allowing the guidance of $\mathcal{G}_b$ to facilitate appropriate context changes according to the new object position, such as new shadows or reflections.

## 4.4. Weighted Guidance with Self-Attention Maps

When applying the gradients of the energy function above, the inaccurate 3D reconstruction and camera paths result in guidance sometimes being applied inaccurately to background regions, for example at incorrect spatial positions. This sometimes results in hallucinated objects in the background regions or other artifacts. To address this, we weight the gradients of the guidance energy using an attention map based on self-attention from the foreground object to other image regions. Intuitively, this includes regions that an edit of the foreground object should affect, including regions that receive updated shadows or reflections, but not regions of the background that should remain unaffected by the edit.

Input Video     Edited Video     Self-Attn. Map

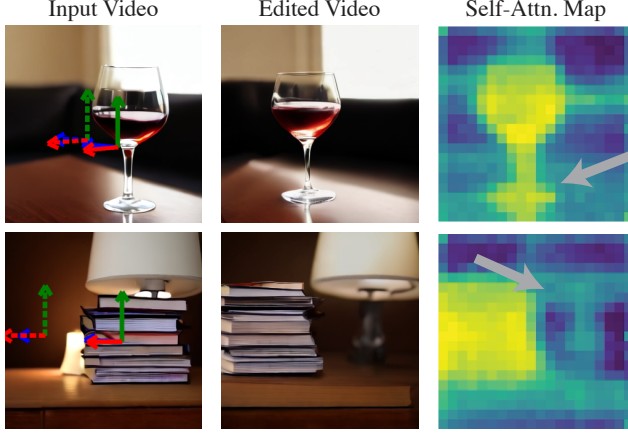

Figure 3. **Visualization of our self-attention-based masks.** The masks do not only include the the edited object, but also regions requiring semantic adjustments, such as a new reflection under the wine glass and newly disoccluded lamp.

We denote the query and key features of the $i$-the frame, stacked across all spatial self-attention layers and flattened as $\boldsymbol{Q}^{(i)}(\boldsymbol{Z}_t) \in \mathbb{R}^{HW \times 1 \times D}$ and $\boldsymbol{K}^{(i)}(\boldsymbol{Z}_t) \in \mathbb{R}^{1 \times HW \times D}$, both with $H$ and $W$ are flattened into a single spatial dimension. Then, we define the spatial self-attention map $A^{(i)}(\boldsymbol{Z}_t) \in \mathbb{R}^{HW \times HW}$ for the $i$-the frame as:

$$\boldsymbol{A}^{(i)}(\boldsymbol{Z}_t) := \boldsymbol{Q}^{(i)}(\boldsymbol{Z}_t)\,\boldsymbol{K}^{(i)}(\boldsymbol{Z}_t), \qquad (15)$$

We then find regions that the transformed object pays attention to by multiplying with the transformed object mask $\boldsymbol{M}_{\text{tgt}}^{(i)}$ and normalizing:

$$\Lambda^{(i)} := \text{norm}_{[0,1]}\left(\boldsymbol{M}_{\text{tgt}}^{(i)}\,\boldsymbol{A}^{(i)}(\boldsymbol{Z}_t)\right), \qquad (16)$$

where $\text{norm}_{[0,1]}$ denotes normalization of the value range to $[0,1]$, $\boldsymbol{M}_{\text{tgt}}^{(i)}$ is flattened to $\mathbb{R}^{1 \times HW}$ and the resulting self-attention-based mask $\Lambda^{(i)}$ is unflattened to $\mathbb{R}^{H \times W}$.

The final masked and aggregated self-attention map $\Lambda \in \mathbb{R}^{M \times H \times W}$ is obtained by stacking $\Lambda^{(i)}$ along the temporal dimension. Figure 3 shows that the target self-attention map locally highlights not only the target position of the selected object but also regions requiring adjustments for context changes, such as areas for a new reflection or the disoccluded lamp.

With this mask, a guided step of our generative process is defined as:

$$\boldsymbol{Z}_{t+\Delta t} = \boldsymbol{Z}_t + \Delta t \cdot v_\theta^\omega + \Lambda \odot \nabla_{\boldsymbol{Z}_t}\left(\rho_o \mathcal{G}_o + \rho_b \mathcal{G}_b\right), \quad (17)$$

where $\rho_o$ and $\rho_b$ are the step sizes for the gradients of $\mathcal{G}_o$ and $\mathcal{G}_b$, respectively.

### 4.5. Null-Text Prediction on Original Object Region

When transforming an object, it is undesirable for the object to remain in its original position while being duplicated in the target position. To avoid this object issue, we employ two techniques. First, at the beginning of the generative process of the target, we randomly initialize the original object area of $\boldsymbol{Z}_0^{\text{src}}$, as highlighted by the source masks $\boldsymbol{M}_{\text{src}}$, and start the generative process from this partially randomized noise. Then, during the generative process, to reduce the influence of text guidance in the original object area and prevent the introduction of a new object in that region, we apply the null-text prediction $v_\theta(\boldsymbol{Z}_t, t, \varnothing)$ within the original object area $\boldsymbol{M}_{\text{src}}$ instead of a prediction with classifier-free guidance [11].

## 5. Experiments

**Dataset.** For quantitative and qualitative comparisons, we generate 27 input videos to be edited, each with a resolution of $320 \times 320$ and 51 frames. To enhance the realism of the generated videos, we lightly finetune OpenSora [51] on 71,556 indoor scene videos from the RealEstate10K dataset [52] for 14,000 iterations.

**Baselines.** In the absence of prior work on modifying 3D object composition in videos, we compare our method to Diffusion Handles [28], the state-of-the-art method for composition editing in 2D images, applying the editing process frame by frame. To further demonstrate the effectiveness of our feature-guided generative process, we also compare it to direct frame warping. Specifically, we first remove the selected object from all frames using an existing inpainting technique [40] and then render the transformed foreground point cloud, $\mathcal{T}\boldsymbol{P}_f$, onto the frames where the selected object is removed. Additionally, we introduce an improved version of the direct frame warping, where the video is further refined using SDEdit [24]. SDEdit is performed for 15 out of the total 30 steps with OpenSora [51].

**Qualitative Results.** Please refer to the supplementary material for the edited video results. We also present snapshots of the edited videos in Figure 1 and Figure 4. Qualitatively, our method successfully edits object composition in videos while making appropriate contextual adjustments, such as the new reflection beneath the wine glass in Figure 1 and the new shadows beneath the transformed car, apple, and vase in rows 2, 3, and 5 of Figure 4, respectively. In comparison, Diffusion Handles [28] (the fourth column in Figure 4) alters the identity of objects or the background across different frames, as seen in the second row, and frequently duplicates objects, as shown in the first row. **These failures are more evident in the videos shown in the supplementary material.** Direct frame warping (the second column) and its refined one by SDEdit [24] (the third

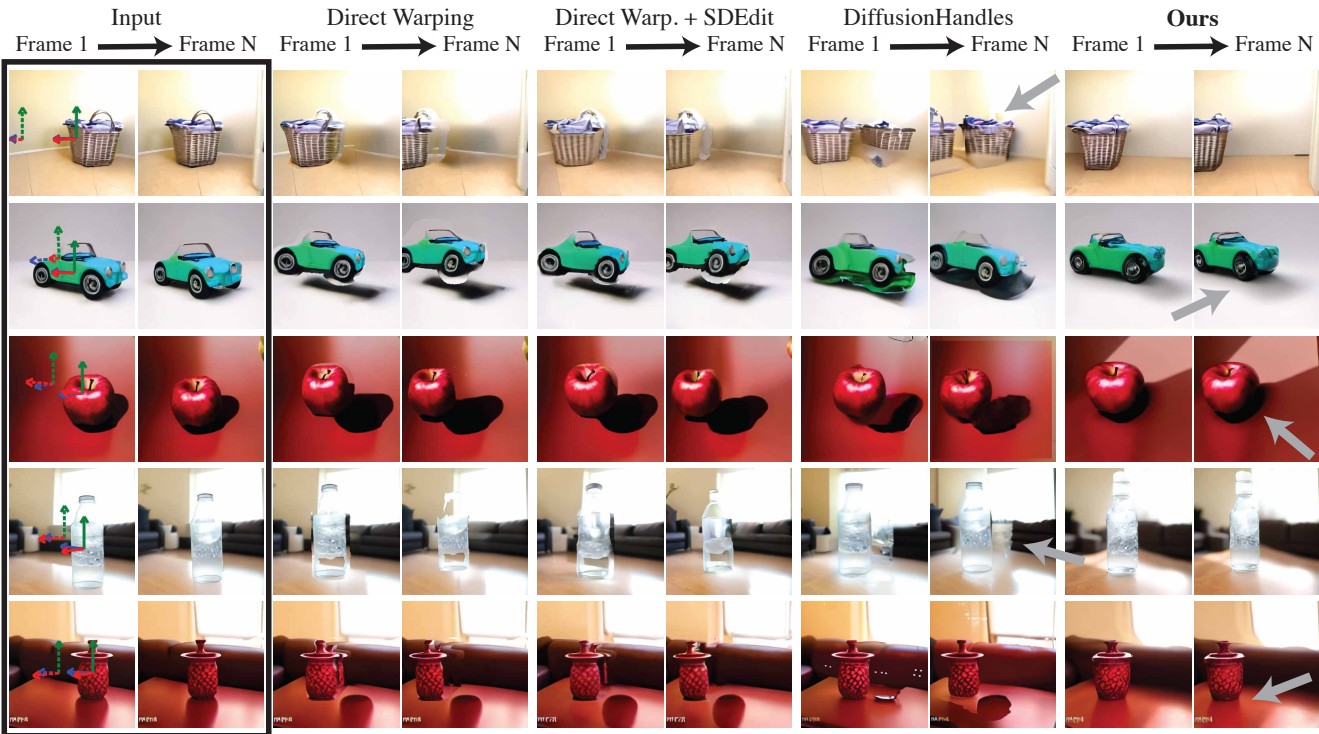

Figure 4. **A qualitative comparison with other baselines.** The examples show that ours best demonstrates plausibility by avoiding object duplication, adjusting shadows properly, and maintaining consistent outputs across frames, desipte warping errors, as illustrated in the direct frame warping outputs (column 2).

column) also typically produce visual seams (second row) and implausible objects (fourth row) due to inaccuracies in warping.

**User Study Results.** Proper quantitative evaluation for video editing results is very challenging, as there are no established metrics for this task. Therefore, we conducted a user study that included questions about the plausibility, identity preservation, and edit coherence of the edited videos. More details about the user study including the queries and setup are provided in the supplementary material. Figure 5 shows human preferences when participants were presented with two videos–one generated by our method and the other by a competing method–along with the input video, and were asked to choose the better one based on each criterion. The results show that our method is preferred over all baselines across all criteria by significant margins. Notably, our method achieved a preference of 100% for plausibility compared to Diffusion Handles [28], and 75% and 57% for identity preservation and edit coherence compared to the SDEdit [24] output of the direct frame warping.

**Temporal Consistency Evaluation.** The biggest advantage of our method compared to per-frame-based editing baselines is its ability to achieve temporal consistency. To

Table 1. **A quantitative evaluation of Frame LPIPS.** Frame LPIPS is scaled by $10^2$, with the best result highlighted in **bold**.

| Per-Frame-Based Editing | | | Ablation Cases | | | Ours |
|---|---|---|---|---|---|---|
| Direct Warp. | Direct Warp. +SDEdit | Diffusion Handles | w/ Temp. Feature | w/o Self-Attn | w/o Null-Text | **Video Handles** |
| 5.19 | 5.03 | 18.63 | 3.81 | 3.77 | 3.79 | **3.71** |

further evaluate this, we introduce a metric called *Frame LPIPS*, which is the average LPIPS [50] score measured between pairs of adjacent frames in the edited video. Frame LPIPS scores for all methods are presented in Table 1. Our method significantly outperforms the baselines, with a score of 3.71 compared to 18.6 for Diffusion Handles [28], demonstrating the superior temporal consistency achieved by leveraging a video prior.

**Ablation Study Results.** We demonstrate the effectiveness of each key aspect of our method through an ablation study involving three cases: using both spatial and temporal self-attention layer features (w/ Temporal Feature, Section 4.2), omitting self-attention-based weighting in the guided generative process (w/o Self-Attn, Section 4.4), and not using null-text prediction in the original object area (w/o Null-Text, Section 4.5). The user study results in the second row of Figure 5 show that our full method outperforms

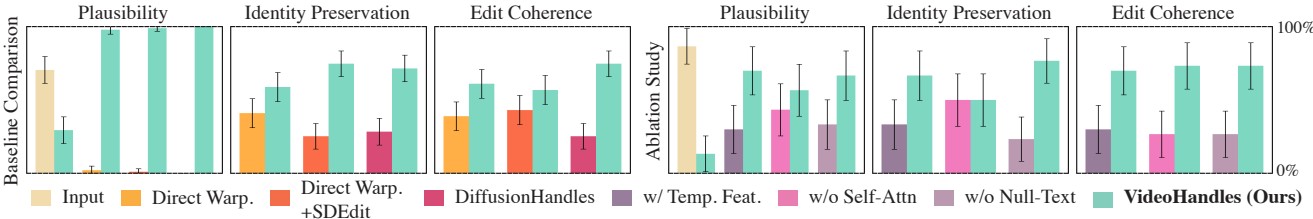

Figure 5. **User study results on the plausibility, identity preservation, and edit coherence of the edited videos.** Each bar pair shows user preferences, with the green bar for our method and the other for the baseline, along with 95% confidence intervals. We also include a comparison with the input video to represent the upper bound of plausibility.

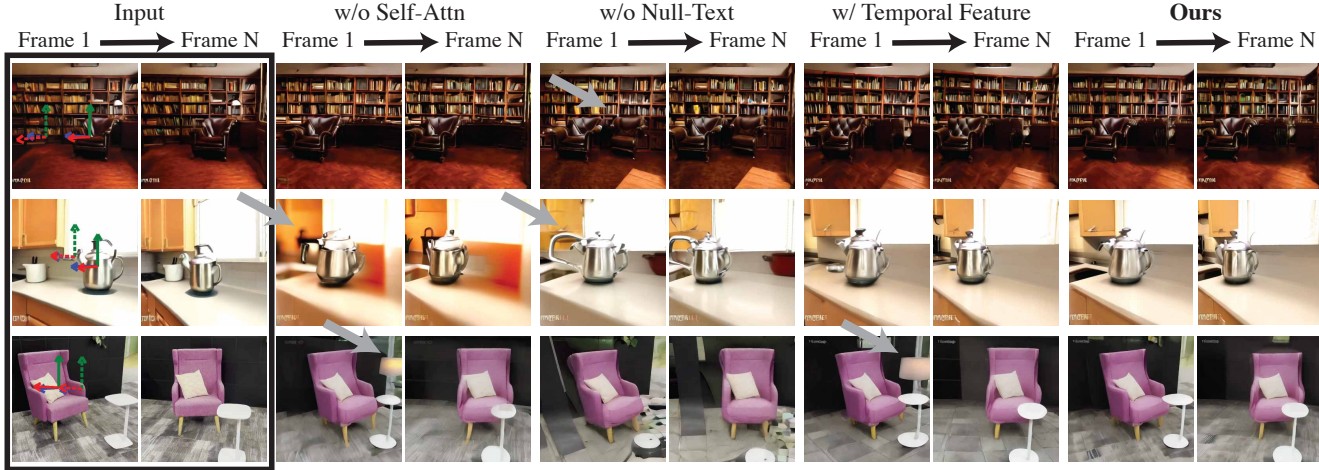

Figure 6. **A qualitative comparison of the ablation study.** We show the effect of each component in our method. As demonstrated, our full method avoids object duplication and unnecessary drastic changes in the background, while effectively preserving the identity of the selected object.

all three cases across all metrics by large margins. Moreover, the best temporal consistency is achieved with our full method, as indicated by the lowest Frame LPIPS score compared to the ablation cases, as shown at the bottom of Table 1. Qualitative comparisons are shown in Figure 6.

Please refer to the supplementary material for the edited video results. In the first row, the results without null-text (third column) exhibit object duplication, showing the armchair in both the original and target positions. In the second row, the results without self-attention-based weighting (second column) drastically alter the background colors, and the results without null-text (third column) introduce a new knob on the kettle. In contrast, our full method (last column) best preserves the identity of the kettle. In the third row, the results without self-attention-based weighting (second column) and with temporal layer features (fourth column) generate a new lamp next to the armchair and thus fail to preserve the background. Our method successfully moves the selected armchair without changing the background.

**Editing Real Videos with Object Composition.** We also showcase the results of editing real videos using our method, as seen in the rightmost image in Figure 1 and the third row of Figure 6. In these examples, the apple in the former and the armchair in the latter are moved to new positions, with shading and shadows generated according to the new composition while successfully preserving the background. To edit the real videos, we mapped the videos to their corresponding random latent noises using the null-text inversion technique introduced by Mokady *et al.* [26].

## 6. Conclusion

We have presented VideoHandles, the first method to our knowledge that leverages the prior of video generative models for editing object composition in video. Given the warping function for each frame obtained from a 3D reconstruction and transformation of an object in 3D space, VideoHandles applies temporally consistent warping to features extracted from a pre-trained video generative model, rather than to the frames themselves, using these features as guidance in the generative process. Experimental results, including a user study, demonstrate that VideoHandles outperforms per-frame editing methods in terms of plausibility, identity preservation, and edit coherence.

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
