# OpenReview forum: "VideoHandles: Editing 3D Object Compositions in Videos Using Video Generative Priors"
_thecvf.com/CVPR/2025/Workshop/SyntaGen — SyntaGen 2025 Poster_

### Official Review · Reviewer_rKAF · 2025-03-27
**The paper presents an interesting research problem and its solution, but the experimental section is still lacking proper evaluation and evidence.**

**Rating:** 5
**Confidence:** 3

**Review:**

Summary

This paper explores generative video editing by transforming an object within a video to a specified 3D spatial location.
To achieve this, they leverage an existing SfM predictor to reconstruct a 3D scene from the input video and construct a pixel-warping function based on a user-provided object mask for generating source point clouds, 3D translation, and rotation parameters. However, instead of editing each video frame by directly warping pixels, which often results in a noisy image and requires post-processing to handle empty areas, they warp encoded pixel-wise latent representations using a video diffusion model. These warped feature maps then serve as conditions for a video diffusion model to generate a seamless video with the object in its new warped position.

Strengths

- The problem setup is interesting and relevant to the workshop.
- The qualitative results are promising, particularly in terms of realistic shadowing, lighting consistency for moved objects, and inpainting at the object's original location, compared to competitors.
- The method is simple yet effective and also shows direct adaptability to OpenSora without requiring additional training.
- The writing is generally clear, and most of the method section is easy to follow.

Weaknesses

- Limited quantitative evaluations: The paper includes only one qualitative comparison. While I understand the lack of established benchmarks or standard evaluation protocols for this task, incorporating photorealistic rendering videos would better simulate test cases aligned with the problem setup.
- Unclear explanation of the weighted self-attention map ($\Lambda$). If I understand correctly, the self-attention matrix $A$ is multiplied by $M_{\text{tgt}}$​, which only includes pixels at the target object position. How does this contribute to generating other elements, such as shadows and background inpainting at the original location (Fig. 3)?
- No supplementary materials were provided despite being mentioned in the paper. Hence, the qualitative results showing temporal consistency of the edited videos and failure cases of competitors could not be checked.
- Rotated object cases are not properly demonstrated. The introduction states that the method supports object rotation, yet no qualitative results illustrate this scenario (only translation was shown).

Conclusion

The problem setup is interesting, and the qualitative results are promising. However, the absence of the promised supplementary materials, limited quantitative evaluation, and unclear methodological explanations weaken the overall contribution. Given these concerns, I assess the paper as below the acceptance threshold.

---

### Official Review · Reviewer_UqLC · 2025-03-28

**Rating:** 6
**Confidence:** 4

**Review:**

**Paper summary**

This work addresses object composition editing in videos according to user input by leveraging a flow-based. The key idea is to warp the model’s features according to the provided 3D transformations and rely on the generative prior of a video diffusion model to produce realistic output videos. To ensure appropriate feature warping, the method introduces a warping function guided by the 3D reconstruction of the video. Several key components are proposed, including Object Transformation Energy, Background Preservation Energy, and a Weighting Guidance Technique. These components help maintain plausible warping, preserve non-edited regions, and allow necessary background changes to account for reflections and shadows. Experimental results demonstrate that this approach outperforms both per-frame image-based editing methods and direct video warping using SDEdit (the closest baseline) in terms of qualitative, quantitative, and user study evaluations.

**Strengths:**
- This work tackles an interesting and novel problem setup.
- This paper works directly with model features, enabling training-free video editing.
- Authors provide a solid baseline comparison, especially in the absence of directly related work.
- The experiments demonstrate high-quality qualitative results, effectively preserving identity while also capturing the movement of shadows and reflections.

**Weaknesses:**
- The contribution of background preservation energy to actual background preservation is unclear, as it only computes an average. Given that the weighting guidance already masks the editable region, is this additional component necessary?
- Since there is no strong baseline for video editing, more results emphasizing temporal consistency would be beneficial. The paper mentions supplementary material, but no such material was provided, making it difficult to assess the video aspects comprehensively.

Despite the absence of the mentioned supplementary material, the results presented in the paper clearly outperform existing methods. Additionally, this work tackles an interesting problem setup and effectively leverages video generation priors, aligning well with the workshop’s scope. I recommend marginally above acceptance threshold, but I strongly encourage the authors to include the supplementary material, as it is crucial for fully assessing the video aspects.

---

### Official Review · Reviewer_TjxZ · 2025-03-28

**Rating:** 7
**Confidence:** 5

**Review:**

The paper introduces a new approach to edit 3D objects using video-based handles. It makes a case for a method of letting users model objects in a video by setting control points, or handles. The approach is based on 3D reconstruction, optical flow estimation, and neural rendering, and ensures realism and temporal consistency in edits across frames. They probably assess the approach qualitatively and quantitatively in comparison to existing methods, achieving greater realism, consistency, or usability.

---

### Decision · Program_Chairs · 2025-03-30

**Decision:**

Accept (Poster)

**Comment:**

The paper received mixed scores of 7 (Accept), 6 (Borderline Accept), and 5 (Borderline Reject). On the good side, the paper tackles an interesting problem setup and effectively leverages video generation priors. The method is simple yet effective, enabling training-free video editing. The qualitative results are promising. On the bad side, some components are unclear, including the explanation of the weighted self-attention map and the contribution of background preservation energy. Quantitative evaluations are limited. Supplementary materials and examples on rotated object cases are missing.

The Program Chair discussed and agreed that the paper's advantages outweighed its shortcomings. Hence, we decided to accept the paper to the workshop. The authors should address the reviewers' concerns in the camera-ready version.